# Influence of chirality and sequence in lysine-rich lipopeptide biosurfactants and micellar model colloid systems

Ian W. Hamley [1] ✉, Anindyasundar Adak[1] & Valeria Castelletto [1]

Lipopeptides can self-assemble into diverse nanostructures which can be programmed to incorporate peptide sequences to achieve a remarkable range of bioactivities. Here, the influence of peptide sequence and chirality on micelle structure and interactions is investigated in a series of lipopeptides bearing two lysine or D-lysine residues and tyrosine or tryptophan residues, attached to a hexadecyl lipid chain. All molecules self-assemble into micelles above a critical micelle concentration (CMC). Small-angle x-ray scattering (SAXS) is used to probe micelle shape and structure from the form factor and to probe inter-micellar interactions via analysis of structure factor. The CMC is obtained consistently from surface tension and electrical conductivity measurements. We introduce a method to obtain the zeta potential from the SAXS structure factor which is in good agreement with directly measured values. Atomistic molecular dynamics simulations provide insights into molecular packing and conformation within the lipopeptide micelles which constitute model self-assembling colloidal systems and biomaterials.

Lipopeptides are a remarkable class of molecule that combine the properties of peptides and lipids conferring biocompatibility and biofunctionality, driving self-assembly and giving biosurfactant properties. Lipopeptides (one class of peptide amphiphile) can self-assemble into different nanostructures depending on their composition as well as the solution conditions[1–8]. Among the nanostructures observed are nanofibrils, nanosheets, nanotubes, vesicles and micelles.

Lipopeptide micelles have also been observed for molecules containing α-helical peptides[9] β-sheet based sequences[10,11] or other short peptide sequences[12–23], conjugates containing intrinsically disordered peptides[24], or cyclic lipopeptides[25–27]. Lipopeptide micelles have a diversity of applications, for example in catalysis. Lipopeptides containing peptides with heme-binding dipeptides (AA, AH, MH or HH) form micelles, which are able to bind carbon monoxide and serve as a model peroxidase[15]. In fact, micelles (self-assembled at pH 7) show superior turnover and catalytic efficiency compared to fibrils that self-assemble from a given lipopeptide at pH 11[15]. In another example, lipopeptide PRW-O-C$_{16}$ bearing an N-terminal proline residue was designed to facilitate biocatalysis via a model aldol reaction[28]. Relevant to light harvesting (artificial photosynthesis), Fry et al. showed that the

chromophore spin state can be controlled in peptide/zinc porphyrin mixtures which can either produce micelle or fibril self-assembled nanostructures[18]. Bioactive peptides such as those containing peptide hormone sequences, for example those for pancreatic polypeptide PYY, are lipidated for therapeutic applications, and lipidated PYY$_{3-36}$ peptides can form spherical micelles or fibrils depending on the nature of the lipid chain and the solution pH[29,30]. Lipopeptides bearing fragments (6- or 8- residues) of the PYY$_{3-36}$ sequence also form spherical micelles[20]. Liraglutide, an important peptide hormone therapeutic (a glucagon-like peptide 1 agonist drug), forms small micelle-like oligomers in a pH-dependent fashion[31]. Ghosh et al. observed a pH-triggered transition from fibrils to spherical micelles in palmitoyl lipopeptides bearing β-sheet favouring sequences. The transition occurs under conditions of pH and ionic strength similar to those of serum suggesting that the reversible and rapid transition could be useful for in vivo drug delivery and imaging applications[10]. Similarly, Jacoby and coworkers propose that a pH-controlled transition from spherical to worm-like micelles observed in their lipopeptides bearing IDP (intrinsically disordered protein) peptide sequences can be used as a shape-driven release system of encapsulated cargo[23].

---

[1]School of Chemistry, Food Biosciences and Pharmacy, University of Reading, Whiteknights, Reading RG6 6AD, UK. ✉e-mail: I.W.Hamley@reading.ac.uk

Due to their immense potential as greener surfactants (potentially more biodegradable and biocompatible than synthetic compounds) for use in cleaning products, there is considerable literature on the surface activity of the class of biosurfactants that are cyclic lipopeptides produced by bacteria, including surfactin, fengycins, putisolvins, iturins and many others[32–42]. However, measurement of the surface tension of designed lipopeptides bearing linear peptide sequences has been less studied. In one example, Zhang et al. have reported the surface tension of lipopeptides comprising a range of $C_{10}$-$C_{14}$ lipid chains and two or three glycine residues along with CMC values obtained from pyrene fluorescence, and they also examined self-assembly[43].

Here, we examine the micellization of four designed lipopeptides bearing lysine-rich tripeptide sequences via detailed small-angle x-ray scattering studies to probe form- and structure-factor effects as well as molecular dynamics (MD) simulations and measurements of surface tension and electrical conductivity. In a recent preliminary report, we presented the synthesis, micelle self-assembly (at low pH) and antimicrobial activity of the four lipopeptides $C_{16}$-YKK, $C_{16}$-WKK, $C_{16}$-Ykk and $C_{16}$-Wkk[44]. Here $C_{16}$ denotes a palmitoyl (hexadecyl) lipid chain, Y, W and K denote tyrosine, tryptophan and lysine respectively and k is D-Lys. We have also investigated hemocompatibility of these molecules as well as presenting further antimicrobial activity data (membrane permeabilization studies), along with a study of self-assembly into fibrils, nanotapes and nanotubes at pH 8[45]. These molecules were designed to incorporate two cationic lysine (or D-lysine) residues to promote self-assembly and to confer antimicrobial activity. The tyrosine and tryptophan residues that link the cationic dipeptide sequences and the lipid chains confer potential aggregation-promoting properties due to π-stacking interactions (and in the case of W also enable fluorescence assaying). Two of the four lipopeptides bear two D-Lys residues since these may influence self-assembly and they also, being non-natural amino acids, confer enhanced biostability (for example, reduced enzymatic degradation in vivo). Here, we demonstrate that detailed SAXS analysis including consideration of structure factor effects which are apparent in high concentration solutions can be used to obtain surface potentials close to experimentally obtained zeta potential values, as well inter-micellar potential energy curves. The form factor analysis provides a wealth of detail on intra-micellar structure. This sheds light onto distinct properties when comparing lipopeptides with either aromatic residue at the N terminus of the peptide or considering the lysine stereochemistry.

Peptide stereochemistry is of practical relevance in the development of analogues of natural sequences bearing D-amino acid residues that enhance biostability, one approach in the development of peptide derivatives for biomedical applications. Peptide stereochemistry can also have a profound influence on self-assembly since molecular chirality can be amplified or translated within chiral self-assembled structures. In one study, the self-assembly of short model amphiphilic peptides $I_3K$ with different chiral sequences was examined[46]. The handedness of the twisted fibrils formed was found to be driven by the stereochemistry of the C-terminal lysine residue. In another example, the stereochemistry in simple dipeptides of phenylalanine and phenylglycine was found to influence the chirality of twisted fibrils formed[47]. In this study, the sequence was also observed to influence the π-stacking and hydrogen bonding extent and pattern. The effect of tripeptide stereochemistry on the aggregation into different structures (hydrogels, crystals) has recently been examined in detail for FFD tripeptides with different D/L-isomer sequences, turn structures being found to promote gelation under suitable conditions[48]. Stereochemistry and aggregation also influence the side chain pKa values (protonation states).

Here, in addition to comprehensive SAXS analysis of micelle structure and interactions, we also present new atomistic molecular dynamics (MD) simulations to probe micelle structure and molecular conformation and intermolecular interactions. This provides important information on the distinct structural and conformational properties of one of the four lipopeptides, $C_{16}$-WKK. This shows

characteristic conformations which influence micelle structure as well as conformation, probed by circular dichroism spectroscopy[44]. MD reveals that both tryptophan-containing peptides sample more extended ordered conformations than the two tyrosine lipopeptides and also have enhanced π-stacking. These effects are more predominant for $C_{16}$-WKK than $C_{16}$-Wkk.

We also present new data on the critical micelle concentration (CMC) from surface tension and electrical conductivity measurements and compare this to previously reported values from fluorescence probe measurements[44]. We also discuss the surface tension properties in comparison to other biosurfactants and conventional surfactants.

## Results

We first consider the structure of the lipopeptide micelles as probed by SAXS and atomistic MD simulations. These reveal that the self-assemblies are model charged colloidal systems since the SAXS data shows highly spherical micelles, with relatively low polydispersity that show clear structure factor effects (increasing with increasing concentration) arising from electrostatic repulsion. The lipopeptides are studied as TFA (trifluoroacetic acid) salts and micelles are charged species due to the lysine residues. The SAXS data can be analysed to provide information on inter-micellar interactions. Complementary to this, atomistic MD simulations shed unique insight into the intra-micellar structure and molecular packing.

The SAXS data measured for a series of concentrations for all four samples is shown in Fig.1. For all four lipopeptides, the structure factor becomes particularly pronounced at higher concentration (1 wt% and above). The scaling of the data by concentration (shown in representative data for **P2D** in Supplementary Fig. 1) shows a pronounced decrease in forward scattering as concentration increases. This is an indicator for repulsive intermolecular interactions[49], to be expected for charged micelles of lipopeptides bearing two lysine residues, which are cationic under the pH 4.6 conditions studied. The position of the first minimum and maximum in the intensity profiles (due to form factor) is slightly concentration dependent, therefore it is not possible to determine the structure factor $S(q)$ by division by the form factor $P(q)$ under the assumption that the total intensity $I(q) = P(q)S(q)$ (valid for isotropic systems of centrosymmetric particles in the monodisperse approximation)[50]. The structure factor was therefore modelled as described below.

The data were fitted using a model core-shell sphere form factor and at higher concentration (here for 0.5 wt% and higher concentration solutions) structure factor was accounted for using the Hayter-Penfold structure factor for charged spherical colloids. The Hayter-Penfold structure factor is obtained from a repulsive potential which takes the form of a screened Coulomb (Yukawa) potential[51]:

$$U(r) = \frac{\pi\varepsilon_0\varepsilon_r\sigma\psi_0^2 \exp[-\kappa(r-\sigma)]}{r}, r > \sigma \qquad (1)$$

This potential acts outside a hard sphere of diameter $\sigma$ (=$2R_{HS}$) In Eq. 1, $\psi_0$ is the surface potential, $\varepsilon_r$ is the dielectric constant of the solvent, $\varepsilon_0$ is the permittivity of free space and $\kappa$ is the Debye-Hückel inverse screening length defined by:

$$\kappa^{-1} = \sqrt{\frac{\varepsilon_r\varepsilon_0 k_B T}{2e^2 I}} \qquad (2)$$

where $I$ is the ionic strength. The surface potential in Eq.1 is given by

$$\psi_0 = \frac{z_{eff}}{\pi\varepsilon_r\varepsilon_0\sigma(2+\kappa\sigma)} \qquad (3)$$

where $z_{eff}$ is the effective charge on the macroion (here lipopeptide micelle).

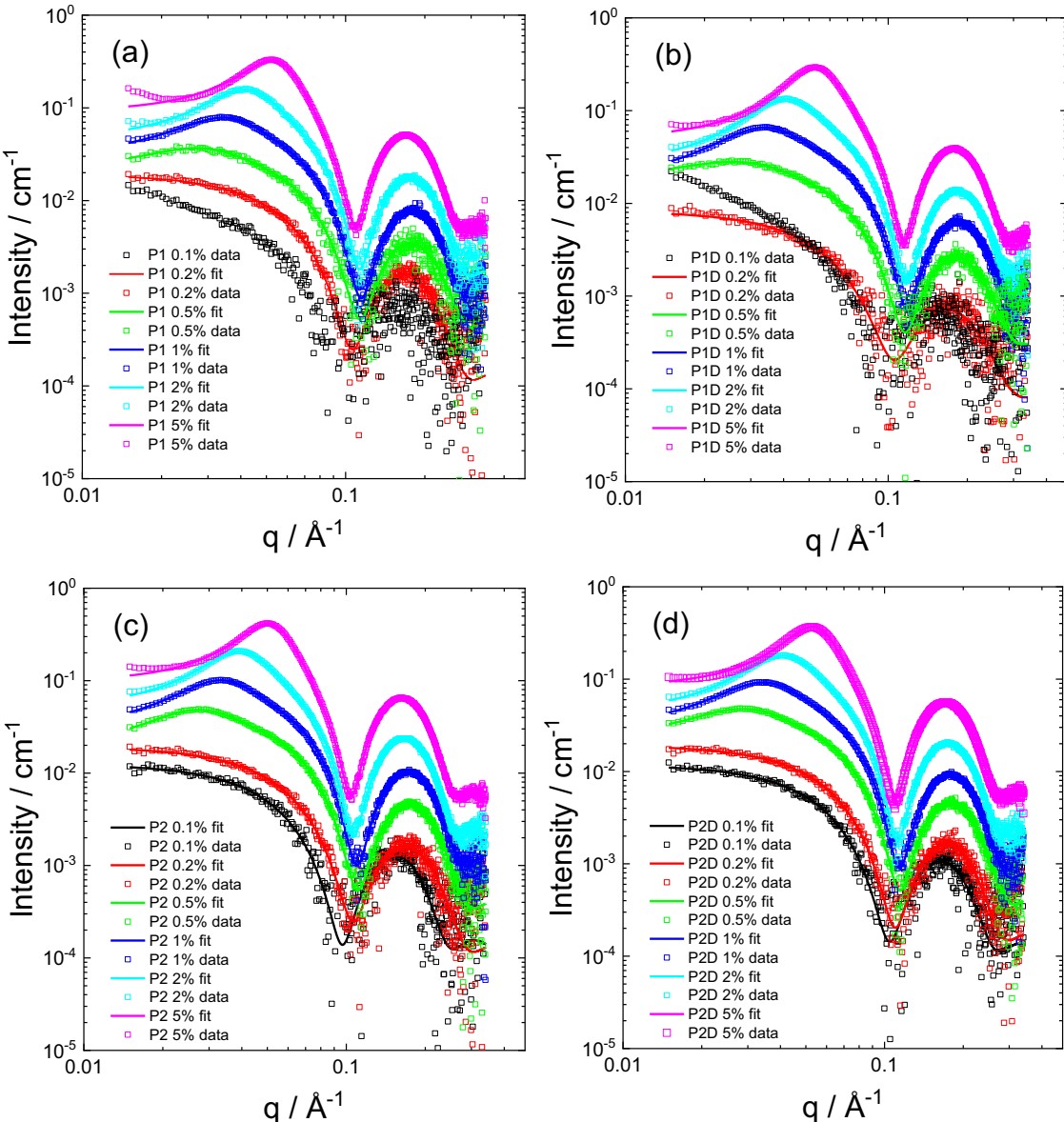

**Fig. 1 | Concentration-dependent SAXS data with model fits.** SAXS data (open symbols) and model fits (lines) measured for (**a**) **P1**, (**b**) **P1D**, (**c**) **P2**, (**d**) **P2D** at the concentrations indicated. For ease of visualization, only every 5th data point is shown.

The Hayter-Penfold structure factor is obtained by analytically solving the Ornstein-Zernike equation in the rescaled mean spherical approximation (RMSA)[51].

The Hayter-Penfold structure factor was found to give high quality fits to the data for the higher concentration solutions studied as shown in Fig. 1. It gives a better fit than the simple hard sphere structure factor, as exemplified by a comparison for the data for 5 wt% **P2D** in Supplementary Fig. 2. In addition, the parameters from a fit using the hard sphere structure factor are unphysical since the volume fraction obtained is too large (above that for hard sphere crystallization or jamming[52,53]) and the associated form factor parameters have a too low value of core radius and a too large relative core/shell scattering contrast factor (Supplementary Table 1).

As evident from Fig. 1 the combination of a core-shell sphere form factor and Hayter-Penfold structure factor provide a good fit to the data for all four lipopeptides over the concentration range studied. It was necessary to account for the variation of form factor parameters

with concentration, since as noted above these change somewhat with concentration (Table 1).

Comparing the SAXS data, one overall trend that is apparent is that **P2** and **P2D** (the tryptophan-containing lipopeptides) have better defined form factor at low concentration than **P1** and **P1D** (the tyrosine-containing lipopeptides) which implies enhanced ordering. Further detailed analysis of the micelle structure is provided by the fit parameters listed in Table 1. Selected parameters for which significant trends with concentration were noted are plotted in Fig. 2.

The values of core radius $R_i$ in Table 1 may be compared to the expected length of an extended $C_{16}$ alkyl chain, $l = 17.6$ Å. The values in Table 1 are in the range 13.5–17.5 Å, lower than $l$ which may indicate either that the chains are not fully extended or that some of the alkyl chain scattering is incorporated in the shell contribution.

Several trends are apparent in the data in Fig. 2. Considering first the form factor data in Fig. 2a, it is evident that $R_o$ increases with concentration for all four lipopeptides, $R_i$ also shows a (less

**Table 1 | Parameters from SAXS data fitting using a core-shell sphere form factor and Hayter-Penfold RMSA structure factor**

|  | 0.1% P1[a] | 0.2% P1 | 0.5% P1 | 1% P1 | 2% P1 | 5% P1 | 0.1% P1D[a] | 0.2% P1D | 0.5% P1D | 1% P1D | 2% P1D | 5% P1D |
|---|---|---|---|---|---|---|---|---|---|---|---|---|
| $R_o \pm \sigma$ / Å | - | 26.51±5.71 | 27.05±4.55 | 26.59±4.42 | 27.77±4.05 | 28.41±3.99 | - | 24.72±6.93 | 26.05±4.70 | 25.83±4.33 | 26.67±4.04 | 27.34±3.85 |
| $R_i$ / Å | - | 14.71 | 14.08 | 14.23 | 14.60 | 15.79 | - | 13.63 | 13.52 | 13.61 | 13.67 | 14.62 |
| $\mu$ | - | -1.761 | -1.614 | -1.506 | -1.460 | -1.241 | - | -2.292 | -1.726 | -1.586 | -1.607 | -1.370 |
| $\eta$ / cm⁻¹ | - | $1.821 \times 10^{-6}$ | $2.913 \times 10^{-6}$ | $4.571 \times 10^{-6}$ | $6.234 \times 10^{-6}$ | $1.012 \times 10^{-5}$ | - | $1.205 \times 10^{-6}$ | $2.653 \times 10^{-6}$ | $4.415 \times 10^{-6}$ | $6.082 \times 10^{-6}$ | $9.848 \times 10^{-6}$ |
| BG | - | $9.376 \times 10^{-5}$ | $3.695 \times 10^{-4}$ | $5.434 \times 10^{-4}$ | $1.266 \times 10^{-3}$ | $3.740 \times 10^{-3}$ | - | $7.453 \times 10^{-5}$ | $2.714 \times 10^{-4}$ | $4.442 \times 10^{-4}$ | $8.963 \times 10^{-4}$ | $2.723 \times 10^{-3}$ |
| $R_{HS}$ / Å | - | - | 72.37 | 53.92 | 46.07 | 40.73 | - | - | 81.10 | 56.63 | 44.39 | 38.85 |
| $\phi$ | - | - | 0.0483 | 0.0522 | 0.0700 | 0.1143 | - | - | 0.0496 | 0.0641 | 0.0639 | 0.1041 |
| $z_{eff}$ | - | - | 14.73 | 27.29 | 34.80 | 168.68 | - | - | 14.64 | 15.26 | 20.72 | 30.37 |
| I / M | - | - | 0.0049 | 0.0116 | 0.0198 | 0.1590 | - | - | 0.0061 | 0.0041 | 0.0072 | 0.0220 |

|  | 0.1% P2 | 0.2% P2 | 0.5% P2 | 1% P2 | 2% P2 | 5% P2 | 0.1% P2D | 0.2% P2D | 0.5% P2D | 1% P2D | 2% P2D | 5% P2D |
|---|---|---|---|---|---|---|---|---|---|---|---|---|
| $R_o \pm \sigma$ / Å | 30.89±4.53 | 26.51±5.71 | 26.72±4.91 | 27.62±4.52 | 28.99±4.25 | 30.26±3.84 | 28.07±4.80 | 26.93±4.74 | 26.38±4.53 | 27.02±4.22 | 28.10±3.91 | 29.61±3.11 |
| $R_i$ / Å | 17.45 | 14.71 | 14.57 | 14.90 | 15.80 | 16.17 | 16.41 | 14.99 | 14.21 | 14.33 | 14.70 | 15.06 |
| $\mu$ | -1.426 | -1.761 | -1.586 | -1.516 | -1.389 | -1.391 | -1.273 | -1.475 | -1.528 | -1.505 | -1.480 | -1.480 |
| $\eta$ / cm⁻¹ | $1.160 \times 10^{-6}$ | $1.821 \times 10^{-6}$ | $3.273 \times 10^{-6}$ | $4.650 \times 10^{-6}$ | $6.353 \times 10^{-6}$ | $9.433 \times 10^{-6}$ | $1.401 \times 10^{-6}$ | $1.936 \times 10^{-6}$ | $3.497 \times 10^{-6}$ | $4.801 \times 10^{-6}$ | $6.496 \times 10^{-6}$ | $9.516 \times 10^{-6}$ |
| BG | $1.023 \times 10^{-4}$ | $9.374 \times 10^{-5}$ | $2.964 \times 10^{-4}$ | $5.855 \times 10^{-4}$ | $1.357 \times 10^{-3}$ | $4.260 \times 10^{-3}$ | $9.066 \times 10^{-5}$ | $1.207 \times 10^{-4}$ | $2.964 \times 10^{-4}$ | $5.641 \times 10^{-4}$ | $1.218 \times 10^{-3}$ | $4.279 \times 10^{-3}$ |
| $R_{HS}$ / Å | - | - | 67.98 | 49.11 | 43.87 | 41.57 | - | - | 65.15 | 49.47 | 43.23 | 45.41 |
| $\phi$ | - | - | 0.0479 | 0.0388 | 0.0518 | 0.1070 | - | - | 0.0451 | 0.0435 | 0.0585 | 0.1616 |
| $z_{eff}$ | - | - | 19.61 | 28.56 | 43.58 | 107.8 | - | - | 18.65 | 21.29 | 24.41 | 63.50 |
| I / M | - | - | 0.0039 | 0.0073 | 0.0154 | 0.0822 | - | - | 0.0055 | 0.0064 | 0.1001 | 0.1000 |

**Key: Form factor:** $R_o$ outer radius ($\sigma_c$ Gaussian polydispersity in c), $R_i$ inner radius, $\mu$ ratio of scattering contrast of inner core/outer core, $\eta$ scattering contrast of core, BG constant background. **Structure factor:** $R_{HS}$ hard sphere radius, $\phi$ Effective volume fraction, $z_{eff}$ effective charge, I ionic strength (temperature fixed at T = 293 K).
aNot possible to obtain reliable fitting parameters.
The monodisperse approximation to factor the form and structure factors of monodisperse isotropic solutions of centrosymmetric particles is assumed.

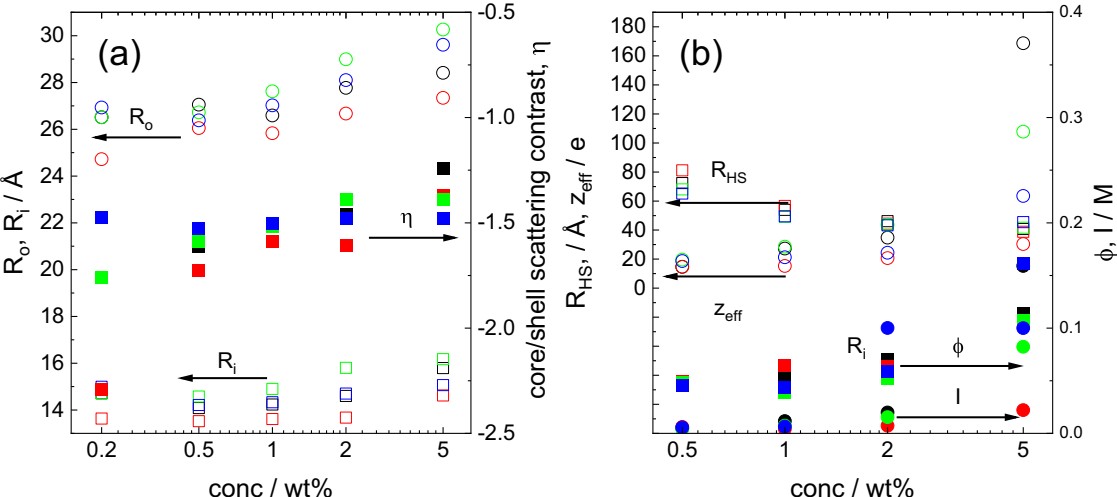

**Fig. 2 | Micelle structure parameters from SAXS data.** Concentration dependence of parameters from SAXS fitting. **a** Micelle structural parameters, Open circles: $R_o$, open squares: $R_i$, closed squares: $\eta$. **b** Structure Factor parameters. Open circles: $z_{eff}$, open squares: $R_{HS}$, closed squares: $\phi$, closed circles: $I$. Colouring: **P1** black, **P1D** red, **P2** green, **P2D** blue.

pronounced) increase with concentration. The core-shell scattering contrast also becomes less negative as concentration increases, i.e. the micelles appear to become more diffuse. The data in Fig. 2a also show that **P2** has notable differences compared to the other samples in having generally larger outer radius $R_o$ and more notably larger inner core radius $R_i$ than for other samples at a given concentration. This was noted in our previous paper[44]. We return below to analyse this difference observed for **P2**, which also shows a different circular dichroism spectrum to the other three samples[44], in terms of conformational properties probed by MD simulations.

Considering the structure factor parameters plotted in Fig. 2b, several trends can be discerned. First, the hard-sphere radius $R_{HS}$ decreases with concentration for all samples, while the effective charge $z_{eff}$ shows little significant difference as a function of concentration (or from sample-to-sample) except at the highest concentration studied, 5 wt%. The ionic strength $I$ shows a slight increasing trend with concentration, while the hard sphere volume fraction $\phi$ also increases with concentration, with a particularly notable enhancement at the highest concentration. There are no significant differences from sample-to-sample in terms of trends evident in the data in Fig. 2b. The values for $z_{eff}$ in Table 1 and Fig. 2b should be considered in the context of the full charge on a micelle bearing 70 molecules (see below) each with two +1 charged lysine (or D-Lys) residues, i.e. + 140, all values in Table 1 being below this value with one exception. The data in Table 1 does indicate that the micelles are highly charged under the conditions studied. The only significant difference in $z_{eff}$ for the four lipopeptide micelle systems is observed at the highest concentration studied and it seems that this quantity is substantially larger for **P1** and **P2** than the corresponding **P1D** and **P2D** lipopeptides. This points to notable influence of chirality on the charge of the lysine residues. An influence of chirality on the distribution of charged species has recently been reported for self-assembling (fibril-forming) FFD tripeptides[48].

The lipopeptide micelles also serve as surfactants as shown by surface tension measurements (Fig. 3). The CMC was determined from the graphs in Fig. 3, as the lowest concentration above which γ reaches a plateau. In this way, values of the CMC $c_{CMC}$ = 0.08, 0.11, 0.04 and 0.04 wt% were determined for **P1, P1D, P2** and **P2D**. The data in Fig. 3 was also used to calculate the surface per molecule head at the air water interface, $A$, the results being displayed in the corresponding

graph. The values are significantly higher than those previously obtained from fluorescence probe measurements using ANS [8-anilo-1-naphthalenesulfonic acid] which are $c_{CMC}$ = 0.00645 ± 0.05 wt % for **P1**, $c_{CMC}$ = 0.00707 ± 0.03 for **P1D**, $c_{CMC}$ = 0.00204 ± 0.04 wt % for **P2**, and $c_{CMC}$ = 0.00257 ± 0.05 wt % for **P2D**[44]. This is discussed shortly.

The CMC was also determined through measurements of electrical conductivity since this is another colligative property that shows a discontinuity at the CMC[53,54]. The concentration-dependent conductivity data for the four lipopeptides is shown in Fig. 4. The conductivity curves show in general a breakpoint very close to the CMC values in Fig. 3, providing confidence in the accuracy of the CMC determination from two independent measurements of distinct colligative properties. The CMC values obtained in the present work are substantially higher than those reported in our previous paper from ANS fluorescence experiments[44]. This is possibly due to the dye molecules influencing the self-assembly process (promoting aggregation at lower concentration) which is avoided with surface tension and conductivity measurements which do not require added probe molecules. It is also possible that the ANS fluorescence measurements are detecting pre-micellar aggregation of oligomers and/or there is self-quenching of ANS fluorescence below the CMC. Regardless, the values of CMC from surface tension and conductivity are in agreement and are believed to be reliable.

Concurrently with the measurements of conducivity, zeta potential values were recorded. These are presented in Supplementary Fig. 3. It is interesting to compare these values with the surface potential from the SAXS data, computed from Eq.3. This leads to a value $\psi_0$ = 37 mV for **P1** at 0.5 wt% concentration, which is the same as the limiting trend value of the ζ-potentials (measured at lower concentration) in Supplementary Fig. 3 (the ζ- potential can be equated to surface potential at the effective radius of the particle[54], here micelle). This demonstrates very satisfactory self-consistency between the SAXS data fitting and independent electrophoretic mobility measurements.

The SAXS structure factor parameters can also be used to compute the inter-micellar potential energy curves via Eq. 1. The resulting potential functions for **P1** are shown in Fig. 5. The curves show an increasing repulsive interaction as inter-micellar separation decreases, the potential energy also increasing with lipopeptide concentration. There is a particularly large increase in repulsive interactions between

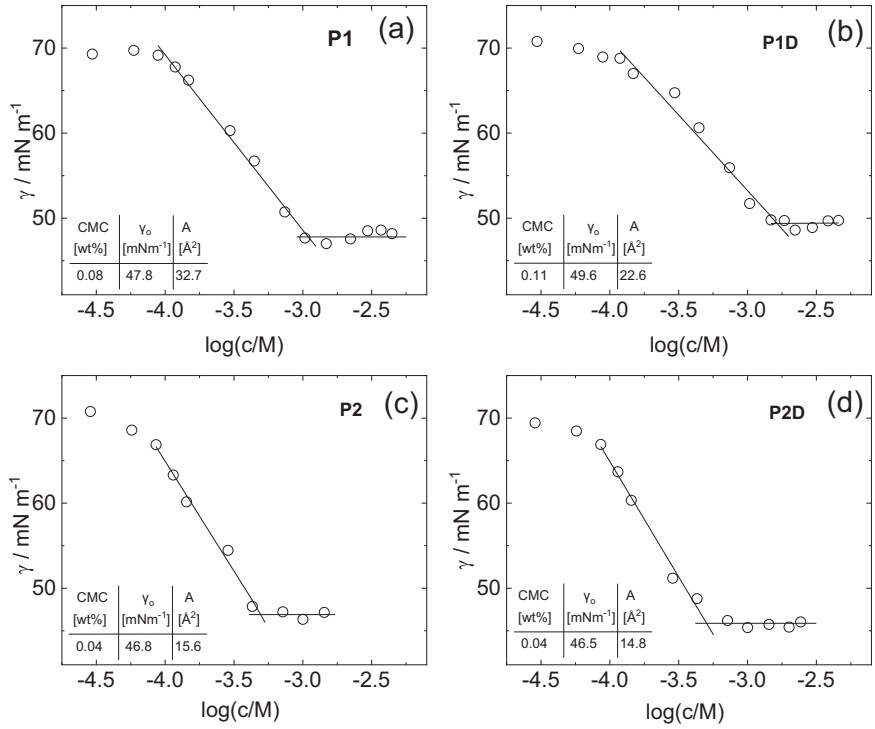

**Fig. 3 | Surface tension data and CMC determination.** Langmuir adsorption isotherms from concentration-dependent surface tension for (**a**) **P1**, (**b**) **P1D**, (**c**) **P2**, (**d**) **P2D**. Inset are values of the CMC from the break points, the limiting surface tension values $\gamma_0$ above the CMC and the areas per molecule $A$ from the slope of the lines below the CMC using the Langmuir adsorption isotherm equation (Eq. 7).

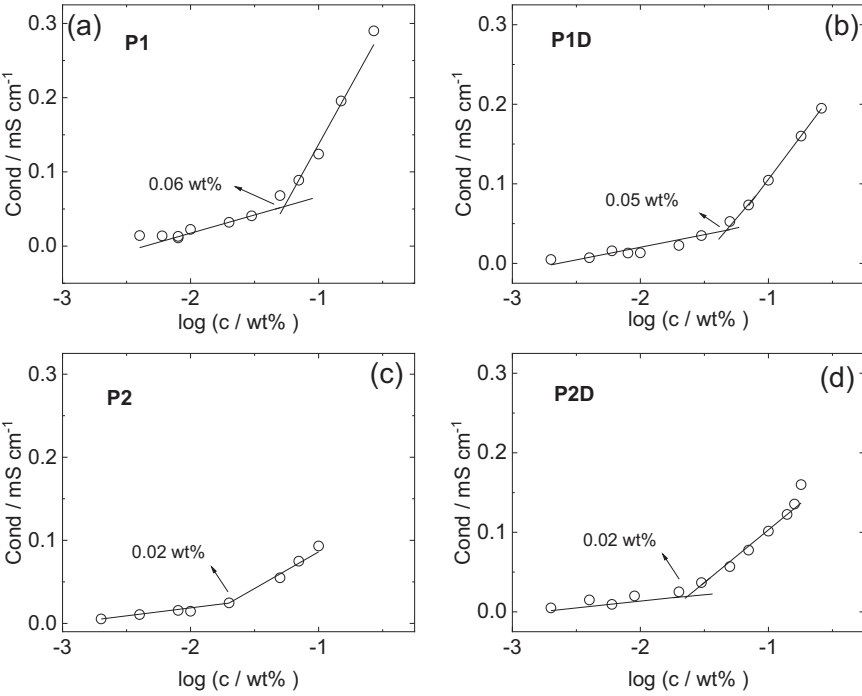

**Fig. 4 | Electrical conductivity data and CMC determination.** Concentration-dependent conductivity data for (**a**) **P1**, (**b**) **P1D**, (**c**) **P2**, (**d**) **P2D** with CMC values indicated.

2 and 5 wt%. The energies in Fig. 5 (at $R_{HS}$, i.e. contact potentials, $U_0$) are in the range of $10–200 k_B T$ except for the 5 wt% values where the potential energy is much larger. The values obtained for the lower concentrations are comparable to those for charged colloidal particles (such as silica beads) with contact potentials $U_0 = 4–17\,k_B T$[55]. It is emphasized that these energies are effective potential energies between micelles and represent a sum of inter-molecular energies. The lipopeptide micelles here are highly charged so higher repulsive potential energies are to be expected. The Debye screening length from Eq. 2 for **P1** at 0.5 wt% is $\kappa^{-1} = 43.1$ Å, smaller values are reported for the silica colloid particles[55] so there is also less charge screening at lower concentrations in our systems (present as TFA salts).

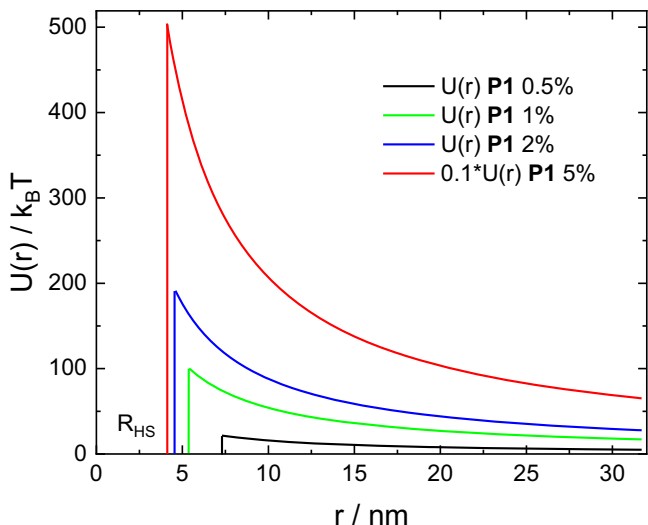

**Fig. 5 | SAXS-derived inter-micellar potential energies.** Potential energies (at 293 K) computed from Eq. 1 using the parameters obtained from SAXS structure factor fitting for **P1**.

Taking **P1** as an example, the area per molecule determined from the surface tension data (using the Langmuir adsorption isotherm equation[53,54]) is $A = 32.7\,\text{Å}$ (Fig. 3). Considering also the calculated surface area of a micelle core $a_{mic} = 4\pi R^2$ (and taking $R = R_i = 14.08\,\text{Å}$ for 0.5 wt% **P1**, Table 1), the association number can be estimated as $p = a_{mic}/A = 76$. Similar values are obtained for the other lipopeptides.

As a further independent check, the association number can be obtained in a model-independent fashion from the forward scattering intensity of the SAXS data[50,56]. The measured SAXS data presented here is in absolute units ($\text{cm}^{-1}$) and the forward scattering (at $q = 0$) can be written as[57]:

$$I(0) = \frac{c_{mic}M_{mic}\left[r_0 v_p(\rho_l - \rho_0)\right]^2}{N_A} \qquad (4)$$

Here $c_{mic}$ is the concentration of micelles, $M_{mic}$ is the micelle molar mass, $r_0$ is the classical electron radius [$0.28179 \times 10^{-12}\,\text{cm e}^{-1}$], $v_p$ is the partial specific volume, and $\rho_l$ and $\rho_0$ represent the lipopeptide and solvent (water) electron density. Here we wish to obtain the micelle molar mass and hence $p$. Rearranging Eq. 4 gives:

$$M_{mic} = \frac{I(0)N_A}{c_{mic}\left[r_0 v_p(\rho_l - \rho_0)\right]^2} \qquad (5)$$

We consider as a representative example **P1**. Using the equation due to Tanford for the volume per lipid chain[58], $v_l = 27.4 + 26.9n$ (where $n$ is the number of carbons in the lipid chain excluding the terminal $CH_3$ group, i.e. $n = 15$ and $v_l$ is in units of $\text{Å}^3$) gives $v_l = 430.9\,\text{Å}^3$. The tail contains 129 electrons in this volume, i.e. the electron density is $\rho_l = 0.299\,\text{e Å}^{-3}$ (which is close to the expected electron density for methylene groups in alkyl chains[59,60]). The electron density of water is taken as $\rho_0 = 0.333\,\text{e Å}^{-3}$. Taking a concentration $c_{mic} = 0.2$ wt% ($0.002\,\text{g cm}^{-3}$) and with $v_p = 259.5\,\text{cm}^3\,\text{mol}^{-1}/225\,\text{g mol}^{-1} = 1.15\,\text{cm}^3\,\text{g}^{-1}$ and using $I(0) = 0.0176\,\text{cm}^{-1}$ (for the data for 0.02 wt% **P1** shown in Fig. 1) leads to $M_{mic} = 4.38 \times 10^4\,\text{g mol}^{-1}$, i.e. $p = M_{mic}/M_{mol} = 65$ [Here $M_{mol} = 675.4\,\text{g mol}^{-1}$ is the molar mass of **P1**[44]]. This is in good agreement with the estimation from the surface tension-derived area/molecule, considering the approximations involved in the use of Eq. 5 in the assumption that the concentration of micelles is equal to the total surfactant concentration and in the estimation of electron density and specific volume.

To probe in more detail the observed differences in the ordering of the lipopeptides within the micelles, atomistic MD simulations were performed. Consistent with the estimated value of $p$ from the surface tension, and the value obtained from the SAXS forward scattering intensity, the MD simulations were carried out for micelles with $p = 70$. This was also found to be reasonable based on calculations of the form factor of micelles with this association number as shown in Supplementary Fig. 4. The profiles were computed using CRYSOL which is part of the ATSAS software[61] (mainly designed for protein SAS data modelling) that performs calculations using the Debye equation based on atomic coordinates (in our generated pdb files), allowing for the scattering effects due to displaced solvent and surface hydration[62–64]. Good agreement is noted between the calculations and data for dilute solutions (0.2 wt%) where structure factor effects are absent. It should also be noted that attempts to simulate micelles with larger association numbers were unsuccessful, since it was not possible to generate sterically acceptable initial packings of molecules for $p = 100$ or more for most of the four lipopeptides (our earlier preliminary value $p = (108 \pm 20)$[44] is therefore considered an over-estimate).

The simulations showed stable micelle structures as exemplified by the images in Fig. 6 obtained for equilibrated configurations after 0.1 ns constant-NPT simulations. Equilibration is confirmed by the observed plateauing of computed properties including SASA (solvent-accessible surface area), the free energy of solvation and the SASA-related volume and density as shown in Supplementary Fig. 5. The simulations provide unique insights into inter-molecular structures. Radial distribution functions (RDFs) were computed for atoms in aromatic residues Tyr (in **P1** and **P1D**) or Trp (in **P2** or **P2D**) as shown in Fig. 7. The RDFs show broad maxima at around 0.55 nm. This peak is enhanced for **P2** and **P2D**, and for **P2** an additional peak is observed at 0.42 nm (Fig. 7), which is associated with enhanced π-stacking interactions in **P2**. Further insight into this was provided by examining the peptide conformations in the micelles via Ramachandran plots (Fig. 8). These reveal significant differences in conformational sampling comparing the two Tyr-containing lipopeptides **P1** and **P1D** and the two Trp-containing ones **P2** and **P2D**. The β-sheet type backbone angles (top left blue area) are notably more populated in the Ramachandran plots for the latter two molecules, while there is also significant sampling of right-handed α-helical conformational states for **P2** (bottom left blue area). This was further interrogated at the residue level. Ramachandran plots for the Tyr residue only (Supplementary Fig. 6) reveal that conformations in the β-sheet region are sampled in preference to the right-handed α-helical region, the corresponding sampling of the latter region apparent in Fig. 7 therefore being assigned to the Lys or D-Lys residues in **P1** or **P1D** respectively. For **P2** and **P2D**, the Trp residues additionally sample left-handed α-helix conformations (top right-hand green box in SI Fig. S6), to a greater extent than do the Tyr residues in **P1** and **P1D**.

The Ramachandran plot analysis shows that **P2** and **P2D** sample more highly ordered conformations and **P2** is particularly distinct from the other three lipopeptides with significant sampling of ordered conformations, both β-sheet (Trp) and right-handed α-helical (Lys or D-Lys). This, combined with the evidence for π-stacking from MD RDF analysis (Fig. 7) is the probable reason for the distinct properties of micelles of **P2** as revealed by the SAXS form factor parameters (Fig. 2a, Table 1) as well as the previously reported[44] distinct circular dichroism spectrum. Although the 230 nm peak in the CD spectrum due to Trp (indole group) is similar in the CD spectra for **P2** and **P2D**[44], the overall shape of the spectra is very different. This reflects distinct conformational as well as π-stacking interactions. The 230 nm peak itself arises from intrinsic electronic transitions in the indole group of Trp[65,66] and is not itself a

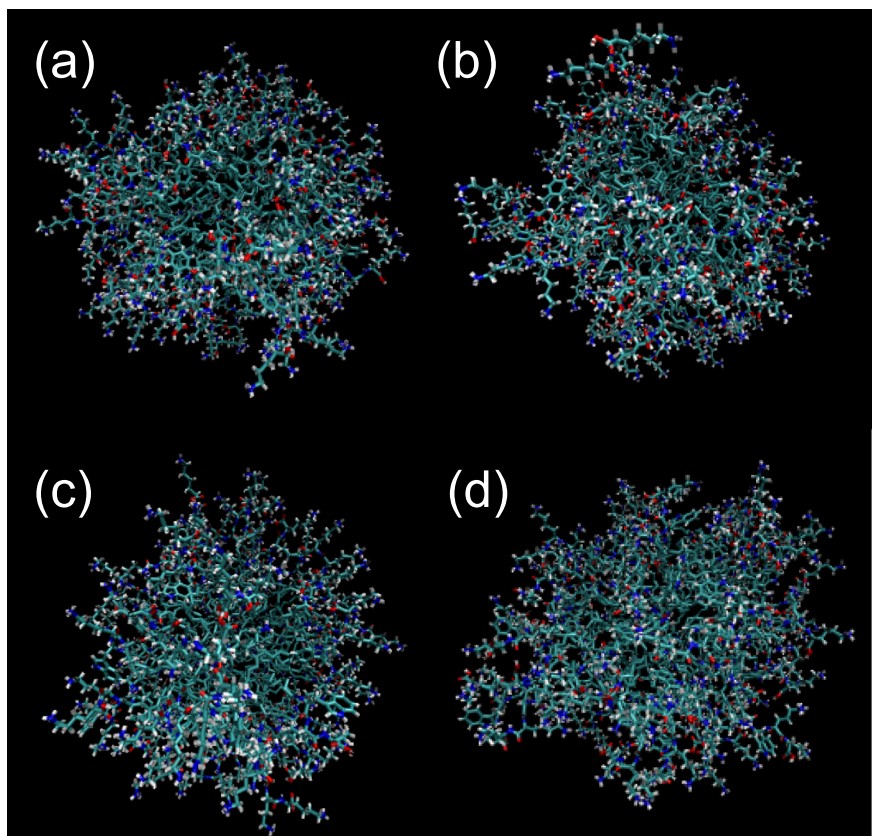

**Fig. 6 | Images of micelle structures from molecular dynamics simulations.** Snapshots (final frame) of micelles from MD simulations. (**a**) **P1**, (**b**) **P1D**, (**c**) **P2**, (**d**) **P2D**.

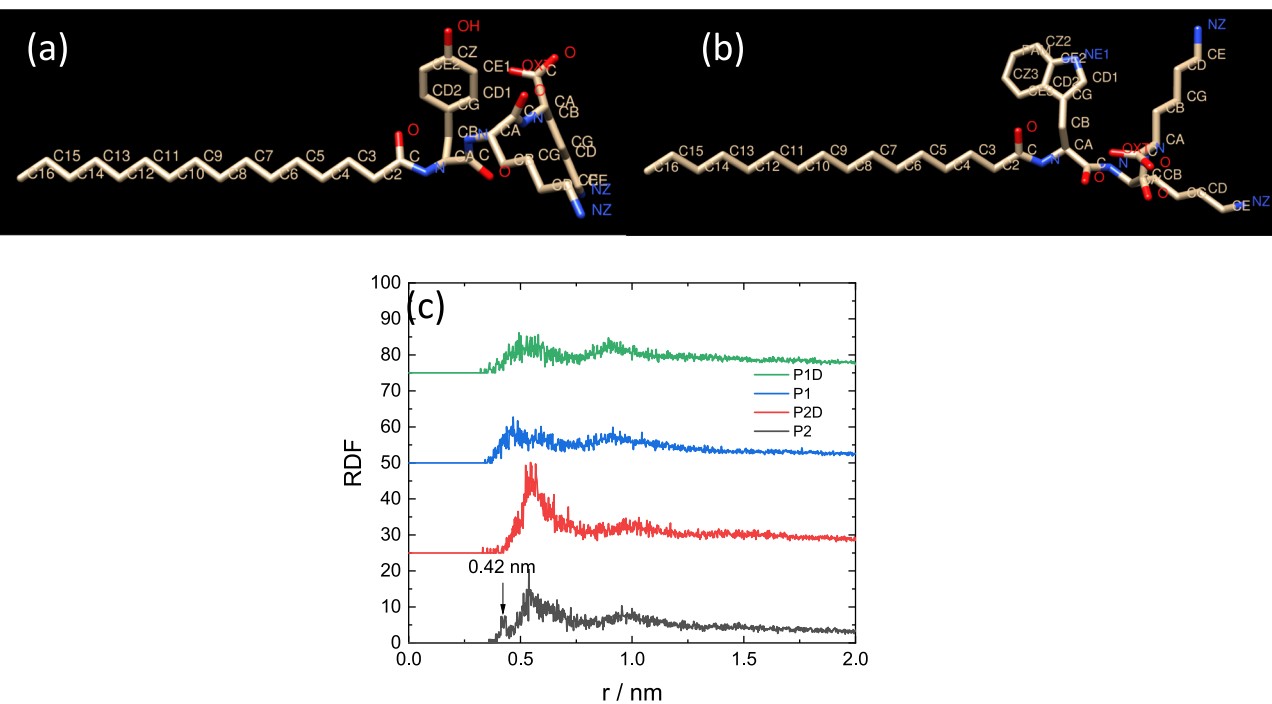

**Fig. 7 | Radial distribution functions (RDFs) from molecular dynamics simulations. a** Atom labelling for **P1**, (**b**) Atom labelling for **P2**, (**c**) RDFs for atoms in aromatic residues (atoms CD2) (top three curves offset for convenience).

signature of π-stacking. Both Trp-containing lipopeptides show a greater degree of peptide backbone ordering. The SAXS data for **P2** point to a more extended conformation, in particular since the shell thickness (comprising the peptide chains) is larger than for the other lipopeptides (Table 1 and Fig. 2a). In this more extended conformation, π-stacking interactions are enhanced, and the combination of these two (coupled) factors may give rise to the distinct CD spectrum.

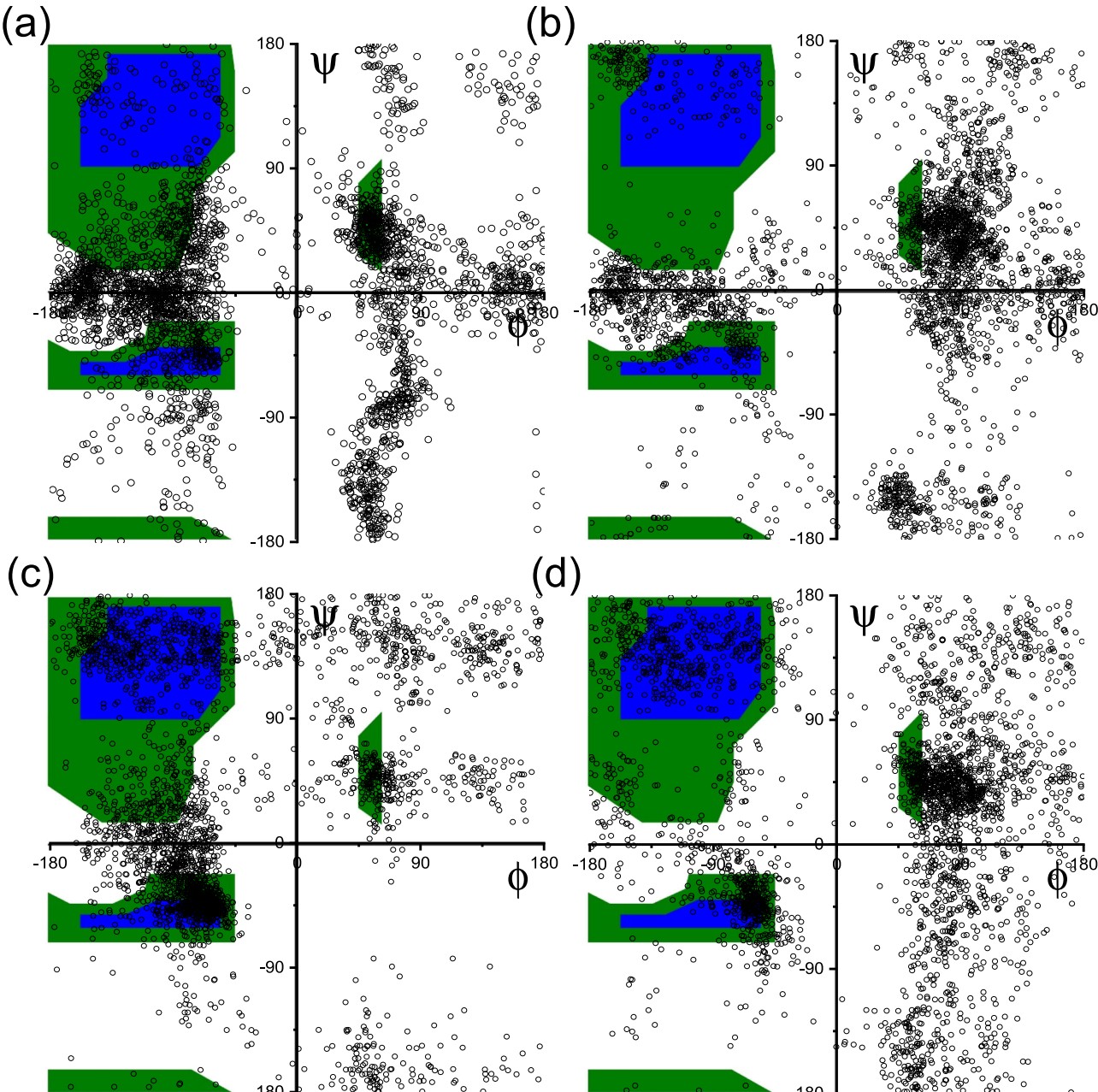

**Fig. 8 | Ramachandran plots from MD simulations.** All-residue Ramachandran plots of all backbone dihedral angles (simulation frames from $t = 900$ ps to $t = 1000$ ps) computed from MD simulations. (**a**) **P1**, (**b**) **P1D**, (**c**) **P2**, (**d**) **P2D**.

## Discussion

In summary, we introduce a systematic study of lipopeptide micelles across multiple length scales from atomic (atomistic MD) through molecular (MD and SAXS) to supra-molecular (micelle structure and inter-micellar interactions).

SAXS measurements for concentration series of lipopeptides have provided insight into micelle structure and inter-micellar interactions through analysis of the structure factor. This has enabled the determination of the inter-micellar potential from experimental data using a model structure factor based on screened Coulombic interactions that is able to account very well for the measured data. The potential energy is an effective overall interaction potential between micelles, i.e. it represents a sum of inter-molecular potentials. Trends in the structure of the micelles as a function of concentration are clear, including the increase in core- and more especially shell- radius with concentration accompanied by reduced core/shell scattering contrast.

Based on SAXS data for glycolipid and phosphocholine lipid micelles, it has been proposed that the position of the secondary maximum in the form factor of micelles can be associated with the shortest center-center separation of surfactant head groups[56]. For the data here the maximum is at $q = (0.172 \pm 0.003)$ Å$^{-1}$ (Fig. 1), corresponding to $d = (36.5 \pm 0.6)$ Å. Taking representative parameters for 0.2 wt% **P1** (Table 1), and considering that the head groups occupy the shell of thickness $R_o - R_i$, we obtain $2\left(R_0 + \frac{(R_0 - R_i)}{2}\right) = 41.2 \pm 8.9$ Å from our model (Table 1), so this estimation appears to be an acceptable first approximation for this lipopeptide (and similarly for the others).

Here we show that it is possible to obtain reliable values of surface potential from SAXS structure factor analysis using the Hayter-Penfold model for charged colloidal (here self-assembling micellar) systems and the values are close to measured ζ-potential values (in slightly more dilute solutions). The method developed could be extended to other systems such as dense colloidal or protein dispersions where

direct ζ-potential measurements may be difficult. We were also able to extract the repulsive inter-micellar potential energy curves from the structure factor. These have rarely been reported for soft materials or peptide-based systems of lipopeptide micelles. The SAXS structure factor analysis also points to a difference in the effective charge on the lipopeptide micelles comparing the L-lysine and D-lysine lipopeptide homologues. This merits further detailed examination such as quantum mechanical modelling.

The limiting surface tension values for the lipopeptides from the data in Fig. 3 lie in the range γ = 45.6–49.6 mN m⁻¹. This is high compared to the limiting surface tensions for bacterial lipopeptides with cyclic peptide headgroups, which are typically in the range γ = 20–30 mN m⁻¹ [32–34,36,38–40,67]. Commercial synthetic surfactants also have low surface tensions, for example for sodium dodecyl sulfate (SDS) γ = 38.5 mN m⁻¹ at room temperature [68,69]. Zhang et al. reported values similar to ours in the range γ = 36.7–44.4 mN m⁻¹ for linear lipopeptides bearing di- or tri- glycine peptides [43]. Low surface tensions γ = 32.4–34.6 mN m⁻¹ were observed for lysine-based surfactants bearing two lipid chains connected by an ε-amino lysine-based linker that form micelles (even lower values were observed for systems with longer alkyl chains that form vesicles) [19].

Atomistic molecular dynamics simulations have provided unique insight into the conformation and inter-molecular interactions in the micelles. Lipopeptide **P2** shows π- stacking features which together with its conformation (which is different to that of the Tyr-based lipopeptides) must cause its distinct circular dichroism spectrum [44] and micelle structure properties revealed by detailed SAXS form factor analysis. Detailed MD analysis of conformation shows that the residue adjacent to the lipid chain plays a key role. The Trp residue in **P2** and **P2D** samples extended β-sheet conformations (and left-handed α-helix to a certain extent) and further influences the conformation of the two adjacent Lys (D-Lys) residues which sample α-helical conformations. This behaviour is in contrast to that observed for the two Tyr-based lipopeptides **P1** and **P1D**. The important role of the N-terminal residue in driving the self-assembly of lipopeptides has previously been noted for β-sheet sequences (which can also form micelles dependent on pH) [11]. This was also noted for lipopeptides bearing blocky or alternating sequences of V and E that form different extended nanostructures (twisted or straight fibrils or nanobelts) [70].

The lipopeptides form well-defined spherical micelles. This is in contrast to many conventional surfactants such as sodium dodecyl sulfate [71] which form rather ill-defined and non-spherical micelles. Together with their interesting structure factor effects, this gives them characteristics of model self-assembling colloidal systems. It should be noted however that the SAXS study here shows that structure factor and form factor effects cannot be decoupled, i.e. the micelle structure is not concentration-independent, which of course is distinct from the behaviour of solid colloidal particles, but is inherent to the self-assembled nature of micelles. The lipopeptide micelles can also be considered as simple models for small charged globular proteins such as lysozyme, which presents cationic residues at its surface and which also has properties such as antimicrobial activity similar to the lipopeptides studied here [44]. The charge of the lipopeptide micelles presented here is significantly higher than that of lysozyme (+ 8 at pH 7 [72]) which could be advantageous in terms of antimicrobial activity. Of course, in contrast to globular proteins which form through peptide chain folding, micelles are non-covalent structures that result from a self-assembly process. It will also be of great interest to extend our approach to examine interactions between micelles in solutions of controlled ionic strength, and to study other types of micellar systems with different morphologies (wormlike or disc-like micelles for example). Finally, a further promising avenue for future research is to investigate the self-assembly of anionic lipopeptides and associated inter-micellar interactions, or to use SAXS with MD to investigate inter-molecular interactions between charged proteins.

In summary, the combination of SAXS and MD along with surface tension/conductivity measurements provides a comprehensive and self-contained analysis of the structure of lipopeptide micelles and their interactions. Future work could include employing the obtained inter-micellar potentials in coarse-grained MD simulations to model inter-micellar interactions based on experimentally-derived data. There is also scope to improve the biosurfactancy properties by tuning the peptide sequence and/or lipid chain length. Here, we have shown that lipopeptides bearing simple cationic tripeptide sequences self-assemble into chiral micelles with sensitivity to sequence and residue stereochemistry.

## Methods
### Materials and sample preparation
**Materials.** Rink amide resin, Fmoc amino acids, diisopropylethylamine (DIPEA), O-(1-benzotriazolyl)−1,1,3,3-tetramethyluronium hexafluorophosphate (HBTU), and triisopropylsilane (TIS) were obtained from Sigma-Aldrich. Methanol, trifluoroacetic acid (TFA), piperidine, diethyl ether, phenol, dichloromethane, and N, N′-dimethylformamide (DMF), HPLC grade water, and HPLC grade acetonitrile were purchased from Thermo-Fisher. HPLC was used to purify lipopeptides using an Agilent 1200 HPLC instrument with a Supelco C-18 column (Zorbax ODS HPLC Column 15 × 4.6 mm, 5 μm) using a gradient of acetonitrile/water at a flow rate of 1.2 ml/min, for 30 min.

**Synthesis of lipopeptides.** The four lipopeptides $C_{16}$-YKK (**P1**), $C_{16}$-Ykk (**P1D**), $C_{16}$-WKK (**P2**) and $C_{16}$-Wkk (**P2D**) (here k denotes D-Lys) were synthesized by solid phase Fmoc peptide synthesis methods [44], using Rink amide resin as solid support. Deprotection of the Fmoc [fluorenylmethoxycarbonyl] groups was achieved using piperidine (20% v/v) in DMF. Each coupling step involved a DMF mixture of Fmoc-amino acids (5 equiv.), DIPEA (12 equiv.) and HBTU (5 equiv.). The reactions for the coupling step and deprotection step were performed by purging nitrogen gas for 6 h and 30 minutes respectively. The free carboxylic group of palmitic acid was coupled at the N-terminus of the synthesized peptide using HBTU and DIPEA in DMF for 6 h. The lipopeptide was cleaved from the resin using a mixture of TFA, TIS, and $H_2O$ [96:2.0:2.0 (v/v/v)], at room temperature for 4 h. Next, TFA was evaporated with nitrogen gas to a minimum volume, and ice-cold diethyl ether was added to yield a white precipitate. The precipitate was centrifuged, lyophilized, and purified by reverse-phase high-performance liquid chromatography (HPLC, Agilent 1200 series). Each synthesized molecule was characterized by HPLC and electrospray ionization-mass spectrometry [44]. The mass values observed ($M_{obs}$) agree with the expected molar masses ($M_{exp}$). The observed values are **P1**: $M_{obs}$ = 675.51 g mol⁻¹ ($M_{exp}$ = 674.51 g mol⁻¹), **P1D**: $M_{obs}$ = 675.52 g mol⁻¹ ($M_{exp}$ = 674.51 g mol⁻¹), **P2**: $M_{obs}$ = 698.53 g mol⁻¹ ($M_{exp}$ = 697.53 g mol⁻¹), **P2D**: $M_{obs}$ = 698.53 g mol⁻¹ ($M_{exp}$ = 697.53 g mol⁻¹). The purities of the lipopeptides analysed by HPLC are as follows: **P1** = 99.62%, **P1D** = 99.70%, **P2** = 99.49%, **P2D** = 96.69%. The yields of the products are as follows: **P1** = 73.94%, **P1D** = 72.28%, **P2** = 76.52%, **P2D** = 75.68%.

### Sample preparation
The samples were prepared by dissolving a measured amount of lipopeptide in a measured amount of water to achieve the desired concentrations in wt%. The pH of these solutions was checked using a Mettler-Toledo Five Easy pH meter with a Sigma-Aldrich micro-pH combination electrode (glass body), which indicated native acidic solutions with a pH of ~4.6. If the pH of any lipopeptide was higher, the pH was adjusted back to 4.6 by adding a few drops of 1 M HCl solution.

### Surface tension
The surface tension γ was measured by the Du Noüy ring method using a Krüss K12 processor tensiometer. A dilution series was prepared from a concentrated stock solution. Aliquots 3 ml of each solution were

placed in a 25 ml beaker, and $\gamma$ was manually measured for each concentration. The solution was left to equilibrate for 10 minutes before each measurement, before $\gamma$ was measured from 3 ring detachments from the surface. The CMC was determined as the concentration at which $\gamma$ reached a stable value ($\gamma_0$). In addition, the slope below the CMC in the plot of $\gamma$ against $\log_{10}(c)$ ($c$: molar concentration), was used to calculate the surface per molecule head at the air-water interface, $A$, according to the Langmuir adorption equation for the surface excess[53,54]:

$$\Gamma = -\frac{1}{RT}\frac{d\gamma}{d[\log c]} \quad (6)$$

Here $R$ is the gas constant and $T$ is the temperature. The surface area per molecule is obtained as

$$A = \frac{1}{\Gamma N_A} \quad (7)$$

Where $N_A$ is Avogadro's number.

### Conductivity and zeta potential
The conductivity and the zeta potential of the solutions were measured using a Zetasizer Nano zs from Malvern Instruments. An aliquot 1 ml of sample was placed inside a disposable folded capillary cell. The sample was left to equilibrate for 120 s before measuring the conductivity, using an applied voltage of 50.0 V. Each measurement presented here is the result of a target 100 measurements.

### Small-angle X-ray scattering (SAXS)
SAXS experiments were performed on beamline B21[73] at Diamond (Harwell, UK). The sample solutions were loaded into the 96-well plate of an EMBL BioSAXS robot and then injected via an automated sample exchanger into a quartz capillary (1.8 mm internal diameter) in the X-ray beam. The quartz capillary was enclosed in a vacuum chamber, to avoid parasitic scattering. After the sample was injected into the capillary and reached the X-ray beam, the flow was stopped during the SAXS data acquisition. Beamline B21 operates with a fixed camera length (3.9 m) and fixed energy (12.4 keV). The images were captured using a PILATUS 2 M detector. Data processing was performed using dedicated beamline software ScÅtter.

### Molecular dynamics simulations
Molecular dynamics simulations were performed using Gromacs[74] (versions 2023.2 and 2020.1-Ubuntu-2020.1-1). Molecules of each of the four lipopeptides were packed using Packmol[75] into spherical micelles with association numbers $p = 70$ or 100 and in some cases additional test values $p = 40$ and $p = 130$. It was found not to be possible to pack molecules of some lipopeptides effectively into larger association number micelles ($p = 100$ and/or $p = 130$) to provide stable configurations for MD and all results here are presented for $p = 70$. The lipopeptide structures were generated using UCSF Chimera with SwissSidechain plugin to add D-lysine residues for **P1D** and **P2D**. Simulations were performed using the CHARMM27 force field[76,77] using the included force field parameters for $C_{16}$ (palmitoyl) chains which were manually adapted to build the lipid-peptide linking unit. The micelles were placed into (9 nm)³ simulation boxes and systems were solvated using spc216 water. Each system was neutralized using a matching number of $Cl^-$ counterions. After energy minimization and 100 ps relaxation stages in the NVT and NPT ensembles, the final simulations were carried out in the NPT ensemble using a leap-frog integrator with 500,000 steps of 2 fs up to 1000 ps. The temperature was maintained at 300 K using the velocity-rescale (modified Berendsen) thermostat[78] with a coupling constant of 10 steps. The pressure was maintained at 1 bar using the Parinello-Rahman barostat[79] and

periodic boundary conditions were applied in all three dimensions. The Particle Mesh Ewald scheme[80,81] was used for long-range electrostatics. Bonds were constrained using the LINCS algorithm[82] and the Verlet cutoff scheme[83] was used. Coulomb and van der Waals cutoffs were 1.0 nm.

## Data availability
The data that support the findings of this study are available from the corresponding author upon request. Source Data files can be accessed via the Figshare repository (https://doi.org/10.6084/m9.figshare.26246933).

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

## Acknowledgements

This work was supported by EPSRC Fellowship grant (reference EP/V053396/1) to IWH. We thank Diamond for beamtime on BM29 (ref. SM34342-1) and Katsuaki Inoue for support.

## Author contributions

Conceptualization: I.W.H.; formal analysis: I.W.H.; funding acquisition: I.W.H.; investigation: I.W.H., A.A. and V.C.; methodology: I.W.H. and V.C.; project administration: I.W.H.; resources: I.W.H.; software: I.W.H.; supervision: I.W.H.; visualization: I.W.H.; writing—original draft: I.W.H.; writing—review and editing: I.W.H.

## Competing interests

The authors declare no competing interests.
