## [Peer Review File · Nature Communications]

Influence of Chirality and Sequence in Lysine-Rich Lipopeptide Biosurfactants and Micellar Model Colloid SystemsREVIEWER COMMENTS

Reviewer #1 (Remarks to the Author):

Hamley et al. report that the chirality of lipopeptides greatly influences the micellar structure and properties. Their findings have a profound impact on applications as biosurfactants and bioactive materials, and are of fundamental importance.

(1) They assumed the decoupling of the structural factor and the form factor. However, this approximation tends to hold well under conditions where the particles do not exhibit significant interactions, or where the systems are ideally monodisperse and the particles are uniformly distributed. Furthermore, the particles should be centro-symmetric. The authors should address this issue before applying it without any annotations.

(2) The specific volume can be measured in a rather simple manner. Why did the authors assume the specific volume?

Minor mistypes

Line 10: "self-self-assemble" => "self-assemble"

Line 397: "Vp" => "vp"

Reviewer #2 (Remarks to the Author):

The manuscript "Lysine-Rich Lipopeptide Micelles: Influence of Chirality and Sequence in Model Colloidal Systems and Biosurfactants" by Hamley, Adak and Castelletto describes a method for investigating the influence of peptide sequence and chirality on micelle structure and interactions among micelles. To this end, four palmitoyl peptides different for sequence and configuration of lysine residues, C16-Tyr-Lys-Lys, C16-Tyr-D-Lys-D-Lys, C16-Thr-Lys-Lys, C16-Thr-D-Lys-D-Lys, were synthesized and used as probes. SAXS measurements as a function of the concentration of the lipopeptides (above CMC) allow obtaining details of the micelle shape and structure from the form factor and of the inter-micellar interactions via analysis of structure factor. Furthermore, the micellar association number and the zeta potential obtained from the SAXS analysis are found in good agreement with the values directly measured. Finally, atomistic MD simulations provide structural features and molecular packing of the lipopeptides in the micelles.

The manuscript reports an original approach, which combines several experimental techniques and MD simulations, to investigate the self-assembling properties of four charged lipopeptides. Importantly, the changes in the peptide sequences allow exploring the effects of chirality and stacking interactions on the packing and conformation of the lipopeptides within the micelles and consequently on their morphology.

The manuscript is clearly written and well organized. It can be published in Nature Comm after minor revisions.

The authors need to state explicitly that the lipopeptides are TFA salts since some readers could not be familiar with solid phase peptide synthesis. This feature is important to understand the origin of Coulombic repulsions, the ionic strength, zeta-potential and inter-micellar potential (the authors refer lipopeptides as TFA salts shortly at line 380).

The authors suggest that the lipopeptide micelles could be considered as models of small, positively charged globular proteins. In this respect, the authors should discuss some questions:

if the same approach could be adopted for the investigation of core-shell nanoparticles with different morphologies;

if it is possible to investigate the effects of the ionic strength due to its importance in determining the screening potential and the Debye length when this parameter is experimentally changed (in the method described in the manuscript, the ionic strength is evaluated by SAXS data);

if negatively charged micelles could be studied adopting the same method being a possible model of negatively charged proteins.

Other points should be discussed.

At fixed concentration, the values of z_{eff} are very different for the four lipopeptide micelles; mainly, large differences are found between P1 and P1D, P2 and P2D. The authors should physically justify these results.

The radial distribution function profile of P2 shows a peak at 0.42 nm. This peak suggests enhanced stacking interactions that would seem absent for P2D. However, a band around 230 nm of similar intensity was observed in the CD spectra of P2 and P2D (data published in reference 44). This band, typical of Trp, suggests that the indole groups are stacked. Experimental CD results and RDF seem not coherent. The author should discuss this point.

Data in Fig. 5 should be reported in kT unit (it would be more meaningful). The temperature value should be indicated in the legend: T=293?

Minors

Line 79: typo "lipeptide"

Line 297: "that" .." should be than

Line 348: "Figure 4" should be Figure 3

Line 422: "were was" should be were

Line 434: "0.1 μ s" should be 0.1 ns

Line 441: Fig. 6 should be Fig. 7

Reviewer #3 (Remarks to the Author):

This manuscript reported experimental and simulation studies on four designed lipopeptides rich in lysine. In general, this is a very neat and standard study of the micellization of lipopeptide using existing experimental and computational approaches. It may be more suitable for a journal of physical chemistry or interfacial phenomena.

Comments:

The authors could provide more insights to justify the importance of lysine-rich lipopeptides. Based on the current manuscript, they seem to be a sub-category of many lipopeptides that could serve as surfactants. What potential functions or unique properties make them stand out?

The author could consider enhancing the connection between the MD simulations and characterization.

How do the water molecules influence the micellization and other properties of the lipopeptides?

The description of simulations may need to be clarified. The authors stated "the final simulations were carried out in the NPT ensemble using a leap-frog integrator with 1,000,000 steps of 2 fs up to 1000 ns". $1,000,000 \times 2 \text{ fs} = 2 \text{ ns}$. Is there a typo here?

Reviewer #1

Hamley et al. report that the chirality of lipopeptides greatly influences the micellar structure and properties. Their findings have a profound impact on applications as biosurfactants and bioactive materials, and are of fundamental importance.

(1) They assumed the decoupling of the structural factor and the form factor. However, this approximation tends to hold well under conditions where the particles do not exhibit significant interactions, or where the systems are ideally monodisperse and the particles are uniformly distributed. Furthermore, the particles should be centro-symmetric. The authors should address this issue before applying it without any annotations.

The reviewer is correct and in addition to the existing text about this on p.10 we have added a further statement in the caption of Table 1 on p.12.

(2) The specific volume can be measured in a rather simple manner. Why did the authors assume the specific volume?

The quantity referred in our manuscript associated with Eq.7 is the specific volume of a lipid chain in a micelle. This cannot be obtained from straightforward experimental measurements and we use an established estimation due to Tanford to estimate it. A specific volume/ density measurement of a micellar solution in water will just give the specific volume of water. The estimation gives a value that provides a self-consistency check based on measured association number.

Minor mistypes

Line 10: "self-self-assemble" => "self-assemble"

Corrected

Line 397: "Vp" => "vp"

Corrected

Reviewer #2

The manuscript "Lysine-Rich Lipopeptide Micelles: Influence of Chirality and Sequence in Model Colloidal Systems and Biosurfactants" by Hamley, Adak and Castelletto describes a method for investigating the influence of peptide sequence and chirality on micelle structure and interactions among micelles. To this end, four palmitoyl peptides different for sequence and configuration of lysine residues, C16-Tyr-Lys-Lys, C16-Tyr-D-Lys-D-Lys, C16-Thr-Lys-Lys, C16-Thr-D-Lys-D-Lys, were synthesized and used as probes. SAXS measurements as a function of the concentration of the lipopeptides (above CMC) allow obtaining details of the micelle shape and structure from the form factor and of the inter-micellar interactions via analysis of structure factor. Furthermore, the micellar association number and the zeta potential obtained from the SAXS analysis are found in

good agreement with the values directly measured. Finally, atomistic MD simulations provide structural features and molecular packing of the lipopeptides in the micelles. The manuscript reports an original approach, which combines several experimental techniques and MD simulations, to investigate the self-assembling properties of four charged lipopeptides. Importantly, the changes in the peptide sequences allow exploring the effects of chirality and stacking interactions on the packing and conformation of the lipopeptides within the micelles and consequently on their morphology. The manuscript is clearly written and well organized. It can be published in Nature Comm after minor revisions.

The authors need to state explicitly that the lipopeptides are TFA salts since some readers could not be familiar with solid phase peptide synthesis. This feature is important to understand the origin of Coulombic repulsions, the ionic strength, zeta-potential and inter-micellar potential (the authors refer lipopeptides as TFA salts shortly at line 380). **This is an important point and the text on p.7 has been changed to emphasize this point right at the start of the manuscript.**

The authors suggest that the lipopeptide micelles could be considered as models of small, positively charged globular proteins. In this respect, the authors should discuss some questions:

if the same approach could be adopted for the investigation of core-shell nanoparticles with different morphologies;
if it is possible to investigate the effects of the ionic strength due to its importance in determining the screening potential and the Debye length when this parameter is experimentally changed (in the method described in the manuscript, the ionic strength is evaluated by SAXS data);
if negatively charged micelles could be studied adopting the same method being a possible model of negatively charged proteins.

The answer to these three questions is respectively: 1) potentially yes; 2) systematic studies of solutions with different ionic strength are also of great interest for future study and 3) it would also be interesting to study negatively charged proteins. We have added a discussion of these points to p.21.

Other points should be discussed.

At fixed concentration, the values of ζ_{eff} are very different for the four lipopeptide micelles; mainly, large differences are found between P1 and P1D, P2 and P2D. The authors should physically justify these results.

This is a very interesting point and indeed the value of ζ_{eff} is notably higher for P1 and P2 compared to P1D and P2D, i.e. the chirality of the lysine residue seems to influence the net charge on the micelles, being significantly lower for the D-lysine based molecules. The text on p.12 and p.19 has been extended (including citation of a recent

relevant reference) to highlight this.

The radial distribution function profile of P2 shows a peak at 0.42 nm. This peak suggests enhanced stacking interactions that would seem absent for P2D. However, a band around 230 nm of similar intensity was observed in the CD spectra of P2 and P2D (data published in reference 44). This band, typical of Trp, suggests that the indole groups are stacked. Experimental CD results and RDF seem not coherent. The author should discuss this point.

Actually, the data in reference 44 show that the CD spectrum of P2 is very distinct from that of P2D although there is some similarity in the intensity of the 230 nm band, in particular the positive maximum at 200 nm observed for P2D is absent for P2. This is explained in part by the differences in π -stacking of indole groups but also the different conformations which are revealed by our MD studies (Fig.7 and Fig.8 and associated discussion), i.e. it is not just distinct π -stacking. Also please consider that the 230 nm band is intrinsic for Trp amino acid itself (see Amdursky, N. and M. M. Stevens (2015). "Circular Dichroism of Amino Acids: Following the Structural Formation of Phenylalanine." *ChemPhysChem* **16**(13): 2768-2774) and is not the result of π -stacking. New text has been added on p.19 to emphasize this.

Data in Fig. 5 should be reported in kT unit (it would be more meaningful). The temperature value should be indicated in the legend: T=293?

This has been done

Minors

Line 79: typo "lipeptide"

Line 297: "that" .." should be than

Line 348: "Figure 4" should be Figure 3

Line 422: "were was" should be were

Line 434: "0.1 μ s" should be 0.1 ns

Line 441: Fig. 6 should be Fig. 7

These have been corrected

Reviewer #3

This manuscript reported experimental and simulation studies on four designed lipopeptides rich in lysine. In general, this is a very neat and standard study of the micellization of lipopeptide using existing experimental and computational approaches. It may be more suitable for a journal of physical chemistry or interfacial phenomena.

Comments:

The authors could provide more insights to justify the importance of lysine-rich lipopeptides. Based on the current manuscript, they seem to be a sub-category of many

lipopeptides that could serve as surfactants. What potential functions or unique properties make them stand out?

These molecules are also valuable as antimicrobials. The text on p.5 has been amended to refer to this in more detail, with reference to an additional submitted manuscript.

The author could consider enhancing the connection between the MD simulations and characterization.

There is a comprehensive cross-comparison of MD simulation and measurements (from SAXS and surface tension) throughout the paper.

How do the water molecules influence the micellization and other properties of the lipopeptides?

We thank the reviewer for this interesting comment. It is well known that micellization is strongly influenced by the hydrophobic effect, i.e. the entropy of water hydrogen bond network disruption although examination of this was not a focus of the present study.

The description of simulations may need to be clarified. The authors stated "the final simulations were carried out in the NPT ensemble using a leap-frog integrator with 1,000,000 steps of 2 fs up to 1000 ns". $1,000,000 \times 2 \text{ fs} = 2 \text{ ns}$. Is there a typo here?

This typo has been corrected

REVIEWER COMMENTS

Reviewer #1 (Remarks to the Author):

The revised manuscript has fully responded to my comments and questions. I strongly support its publication.

Reviewer #2 (Remarks to the Author):

The manuscript "Lysine-Rich Lipopeptide Micelles: Influence of Chirality and Sequence in Model Colloidal Systems and Biosurfactants" by Hamley, Adak and Castelletto was revised according to the suggestions of the reviewers.

The manuscript can be published in Nature Comm.

A typo is present at line 409: conformational should be conformations

Reviewer #3 (Remarks to the Author):

The authors addressed most of the questions raised by the reviewers well. However, it would be helpful if the authors address several major issues about their computation work.

1. The author stated "The micelles were placed into 9 nm³ simulation boxes and systems were solvated using spc216 water". Please check if there is a typo here. 9 nm³ box is a very small box with probably ~2.x nm at each direction. The micelle is very likely to interact with its own mirror in such a small box if standard periodic boundary condition is applied. Please check if this is another typo.

2. The author fixed the typo to state that they collected the data from 1000 ps (1 ns) MD simulation. However, such extremely short simulations could bring an severe artificial effect on the simulation result because the micelle may not have enough time to vary its conformation. Also, analysis based on the last frame of a MD simulation is very likely to present bias data (like Fig. 8). Researchers will need to analyze at least 1000-2000 frames using a statistical manner to obtain convincing data. I will encourage the authors to extend their simulations to at least in a scale of 10-100 ns and analyze multiple frames to reduce the potential bias.

Reviewer #1

The revised manuscript has fully responded to my comments and questions. I strongly support its publication.

Reviewer #2

The manuscript "Lysine-Rich Lipopeptide Micelles: Influence of Chirality and Sequence in Model Colloidal Systems and Biosurfactants" by Hamley, Adak and Castelletto was revised according to the suggestions of the reviewers.

The manuscript can be published in Nature Comm.

A typo is present at line 409: conformational should be conformations

This has been corrected

Reviewer #3

The authors addressed most of the questions raised by the reviewers well. However, it would be helpful if the authors address several major issues about their computation work.

1. The author stated "The micelles were placed into 9 nm³ simulation boxes and systems were solvated using spc216 water". Please check if there is a typo here. 9 nm³ box is a very small box with probably ~2.x nm at each direction. The micelle is very likely to interact with its own mirror in such a small box if standard periodic boundary condition is applied. Please check if this is another typo.

This has been corrected

2. The author fixed the typo to state that they collected the data from 1000 ps (1 ns) MD simulation. However, such extremely short simulations could bring an severe artificial effect on the simulation result because the micelle may not have enough time to vary its conformation. Also, analysis based on the last frame of a MD simulation is very likely to present bias data (like Fig. 8). Researchers will need to analyze at least 1000-2000 frames using a statistical manner to obtain convincing data. I will encourage the authors to extend their simulations to at least in a scale of 10-100 ns and analyze multiple frames to reduce the potential bias.

Equilibration was observed within 1 ns in the MD simulations. This is apparent from the plateauing of all four quantities (SASA, ΔG , density, volume) in Fig.S5. Also other quantities (not shown) were monitored and shown to rapidly plateau, for example the overall radius of gyration (example plotted below). Actually, in our experience, micellar systems stabilize rapidly in MD simulations, after suitable NVT and NPT equilibrations, for instance see our recent paper [10.1016/j.colsurfa.2024.134394](https://doi.org/10.1016/j.colsurfa.2024.134394) and others cited therein

Fig. Radius of gyration and components for **P1**.

The point about statistical averaging is a reasonable one. Therefore the data in Fig.8 and SI Fig.6 have been replaced by Ramachandran plots averaged over 900-1000 ps. The trends are the same as those already discussed, although the plots are a more cluttered with data points (the previous presentation is probably clearer and can be reverted to).

REVIEWERS' COMMENTS

Reviewer #3 (Remarks to the Author):

The authors have addressed the comments. Their experimental observation should provide adequate information to help their simulation results.